# MST: Masked Self-Supervised Transformer for Visual Representation

**Zhaowen Li**[†**]   **Zhiyang Chen**[†**]   **Fan Yang**[°]   **Wei Li**[°]   **Yousong Zhu**[†]
**Chaoyang Zhao**[†]   **Rui Deng**[°▽]   **Liwei Wu**[°]   **Rui Zhao**[°]   **Ming Tang**[†]
**Jinqiao Wang**[†⋆]
[†]National Laboratory of Pattern Recognition, Institute of Automation, CAS
[⋆]School of Artificial Intelligence, University of Chinese Academy of Sciences
[°]SenseTime Research
[▽]University of California, Los Angeles
{zhaowen.li,zhiyang.chen,yousong.zhu,chaoyang.zhao}@nlpr.ia.ac.cn
{tangm,jqwang}@nlpr.ia.ac.cn
{yangfan1,liwei1,dengrui,wuliwei,zhaorui}@sensetime.com

## Abstract

Transformer has been widely used for self-supervised pre-training in Natural Language Processing (NLP) and achieved great success. However, it has not been fully explored in visual self-supervised learning. Meanwhile, previous methods only consider the high-level feature and learning representation from a global perspective, which may fail to transfer to the downstream dense prediction tasks focusing on local features. In this paper, we present a novel Masked Self-supervised Transformer approach named MST, which can explicitly capture the local context of an image while preserving the global semantic information. Specifically, inspired by the Masked Language Modeling (MLM) in NLP, we propose a masked token strategy based on the multi-head self-attention map, which dynamically masks some tokens of local patches without damaging the crucial structure for self-supervised learning. More importantly, the masked tokens together with the remaining tokens are further recovered by a global image decoder, which preserves the spatial information of the image and is more friendly to the downstream dense prediction tasks. The experiments on multiple datasets demonstrate the effectiveness and generality of the proposed method. For instance, MST achieves Top-1 accuracy of 76.9% with DeiT-S only using 300-epoch pre-training by linear evaluation, which outperforms supervised methods with the same epoch by 0.4% and its comparable variant DINO by 1.0%. For dense prediction tasks, MST also achieves 42.7% mAP on MS COCO object detection and 74.04% mIoU on Cityscapes segmentation only with 100-epoch pre-training.

## 1   Introduction

As Yann LeCun said, "if intelligence is a cake, the bulk of the cake is unsupervised learning". This sentence reflects that *Un-/Self-supervised Learning* played a central role in the resurgence of deep learning. Common approaches focus on designing different pretext tasks [10, 29, 14, 3, 4, 6, 5, 13, 1, 36] and aim to learn useful representations of the input data without relying on human annotations. It then uses those representations in downstream tasks, such as image classification, objection detection, and semantic segmentation.

---

[*]Work done as an intern at SenseTime Research.

35th Conference on Neural Information Processing Systems (NeurIPS 2021).

In computer vision, previous methods focus on designing different pretext tasks. One of the most promising directions among them is contrastive learning/instance discrimination [17, 23], which regards each instance in the training dataset as a single category. Based on instance discrimination [14, 4, 6, 5, 13, 1], some methods show the effectiveness in the image classification task. They successfully bridge the performance gap between self-supervised and full-supervised methods. However, almost all of self-supervised learning methods, which formulate the learning as image-level prediction using global features, are suboptimal in the pixel-level predictions [14, 1, 13], such as object detection and semantic segmentation. Also, InfoMin [35] finds that high-level features do not truly matter in transferring to dense prediction tasks. Here, current self-supervised learning may overfit to image classification while not being well tamed for downstream tasks requiring dense prediction.

Meanwhile, large-scale pre-trained models have become the prevailing formula for a wide variety of Natural Language Processing (NLP) tasks due to its impressive empirical performance. These models typically abstract semantic information from massive unlabeled corpora in a self-supervised manner. The Masked Language Modeling (MLM) [10] has been widely utilized as the objective for pre-training language models. In the MLM setup, a certain percentage of tokens within the input sentence are randomly masked, and the objective is to predict the original information of the masked tokens based only on its context. In NLP tasks, we found that the different mask strategies used in the MLM framework had a great impact on the performance of the model. However, in the field of vision, images have higher-dimensional, noisy, and redundant format compared to text. The main information of input images is randomly distributed in tokens. If tokens are randomly masked, it will lead to poor performance. Some of previous methods use random tokens, such as iGPT [3] and ViT [11]. iGPT trains self-supervised Transformers using an amount of 6801M parameters and achieves 72.0% Top-1 accuracy on ImageNet by masking and reconstructing pixels, while ViT trains ViT-B model on the JFT-300M dataset, and the result is significantly lower than the supervised model.

The random MLM is prone to mask the tokens of crucial region for images, resulting in misunderstanding, and is not suitable for directly applying to self-supervised vision Transformers. In order to avoid masking the tokens of crucial region, we propose a masked token strategy based on the multi-head self-attention map, which dynamically masks some tokens of patches without damaging the crucial structure for self-supervised learning. Notably, the strategy would not increase the training time. Also, predicting original tokens alone may cause the model to over-emphasize local region, and therefore suppress the ability to recognize objects. Hence, in this paper, we present a novel Masked Self-supervised Transformer approach named MST, which can explicitly capture the local context of an image while preserving the global semantic information. In addition, a global image decoder is further exploited to recover the spatial information of the image and is thus more friendly to the downstream dense prediction tasks.

We validate our method on multiple visual tasks. In particular, on the ImageNet linear evaluation protocol, we reach 76.9% top-1 accuracy with DeiT-S and achieve the state-of-the-art performance. Overall, we make the following contributions:

- We propose a new masked self-supervised transformer approach called MST. It makes full use of self-attention map to guide the masking of local patches, thus enhancing the understanding of local context semantics in pre-training without damaging the crucial structure.

- Our method can effectively recover the spatial information of the image by a global image decoder, which is vital for the downstream dense prediction task and greatly improves the versatility and scalability of the pre-training model.

- Extensive experiments demonstrate the effectiveness and transfer ability of our method. Specifically, the results on ImageNet [9], MS COCO [18] and Cityscapes [8] show that our method outperforms previous state-of-the-art methods.

## 2 Related Works

### 2.1 Self-supervised visual representation learning

Following MLM paradigm in NLP [10, 25], iGPT [3] trains self-supervised Transformers by masking and reconstructing pixels, while ViT [11] masks and reconstructs patches. Recently, the most competitive pretext task for self-supervised visual representation learning is instance discrimination

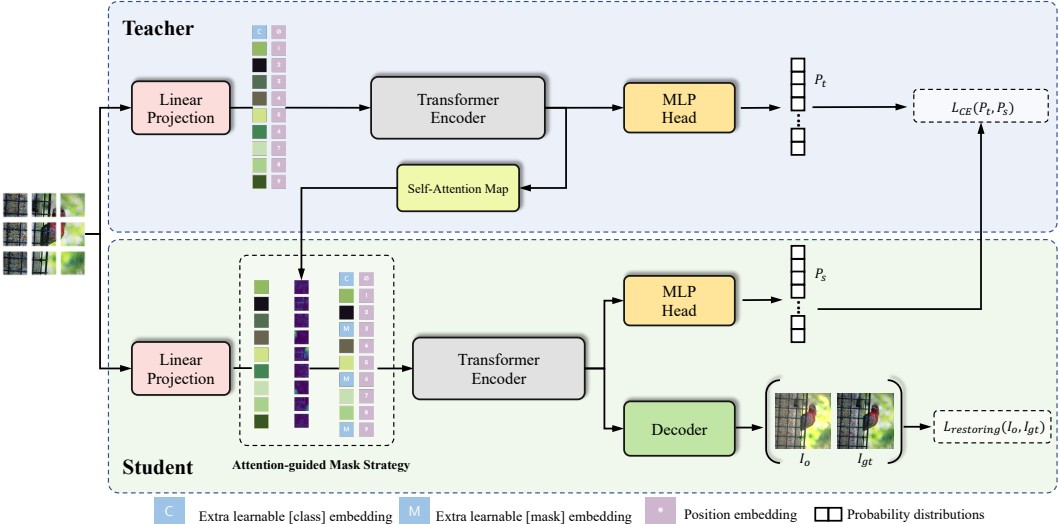

Figure 1: The pipeline of our MST. Both student and teacher share the same architecture with different parameters. Inspired by the MLM in NLP, the attention-guided mask strategy is first introduced to mask the tokens of the student network based on the output self-attention map of the teacher network. The basic principle is to mask some patches with low responses and does not destroying the important foreground regions. Then, a global image decoder is used to reconstruct the original image based on the masked and unmasked tokens. Finally, the total loss function consists of the self-supervised cross entropy loss and the restoring loss.

[14, 4, 6, 5, 13, 1]. The learning objective is simply to learn representations by distinguishing each image from others, and this approach is quite intractable for large-scale datasets. MoCo [14] improves the training of instance discrimination methods by storing representations from a momentum encoder instead of the trained network. SimCLR [4] shows that the memory bank can be entirely replaced with the elements from the same batch if the batch is large enough. In order to avoid comparing every pair of images and incur overfitting, BYOL [13] directly bootstraps the representations by attracting the different features from the same instance. SwAV [1] maps the image features to a set of trainable prototype vectors and proposes multi-crop data augmentation for self-supervised learning to increase the number of views of an image. MoCov3 [7] and DINO [2] apply the self-supervised learning methods of computer vision to Transformers and achieve superior performance in image classification task. These works achieve comparable results compared to supervised ImageNet [9] pre-training. The success of these methods suggest that it is of central importance to learn invariant features by matching positive samples. However, almost all of these self-supervised learning methods formulate the learning process as image-level prediction using global features, so they lack the ability to pay attention to local features.

## 2.2 Self-supervised dense prediction learning

Based on the existing instance discrimination, some researchers propose self-supervised dense prediction methods. Self-EMD [19] adopts Earth Mover's Distance (EMD) to compute the similarity between two embedding. Insloc [33] pastes image instances at various locations and scales onto background images. The pretext task is to predict the instance category given the composited images as well as the foreground bounding boxes. PixPro [31] directly applies contrastive learning at the pixel level. DenseCL [28] presents dense contrastive learning by optimizing a pairwise contrastive loss at the pixel level between two views of input images. VADeR [24] and FlowE [32] also learn dense image representations for downstream detection and segmentation tasks. Meanwhile, HED [34] focuses on downstream segmentation task. These methods also show the effectiveness in detection and segmentation tasks but get poor performance on image classification tasks. In a word, these methods overfit a single task and cannot train a general pre-training model.

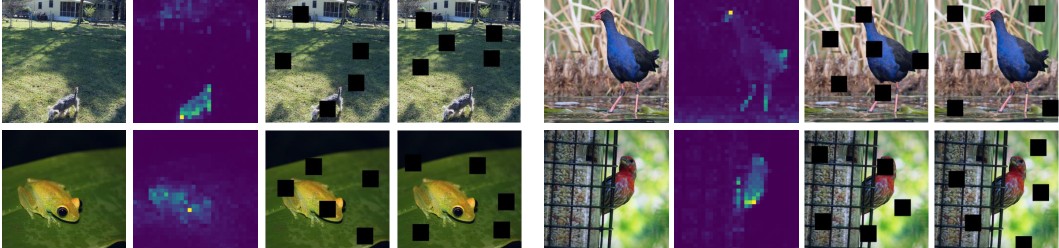

Figure 2: Illustration of our attention-guided mask strategy. It improves by preserving key patterns in images, compared with the original random mask. Description of images from left to right: (a) the input image, (b) attention map obtained by self-attention module, (c) random mask strategy which may cause loss of crucial features, (d) our attention-guided mask strategy that only masks nonessential regions. In fact, the masked strategy is to mask tokens.

## 3 Methods

The pipeline of our proposed MST is shown in Figure 1. We propose a Masked Self-supervised Transformer (MST) approach, which creatively introduces attention-guided mask strategy and uses it to complete image restoration task. Our method is combined with some classical components of instance discrimination, such as the momentum design, asymmetric data augmentations, and multi-crop strategies. Here, we first review the basic instance discrimination method in 3.1. Then, the mechanism and effect of our attention-guided mask strategy are explained in 3.2. Finally, we describe the reconstruction branch and the training target of our method in 3.3.

### 3.1 The basic instance discrimination method

As noted in prior works[4, 14, 13, 29, 1], many existing augmentation policies adopt random resized cropping, horizontal flipping, color jittering and so on. We generate multiple views for each image $x$ under random data augmentation according to multi-crop [1]. This operation can acquire two standard resolution crops $x_1$ and $x_2$ representing the global view and sample $N$ low-resolution crops indicating partial view. They are encoded by two encoders, teacher network $f_t$ and student network $f_s$, parameterized by $\theta_t$ and $\theta_s$ respectively, and outputting vectors $O_t$ and $O_s$. Both encoder $f_s$ and $f_t$ consist of a Transformer backbone and a projection head [5], which share the same architecture with different parameters. The parameters $\theta_t$ of fixed encoder $f_t$ is updated by the moving-average of $\theta_s$ according to Eq (1).

$$\theta_t = m * \theta_t + (1 - m) * \theta_s \tag{1}$$

Given a fixed teacher network $f_t$, the student network $f_s$ learns the parameters $\theta_s$ by minimizing cross entropy loos as Eq (2).

$$L_{CE}(\theta_s) = \sum_{i \in \{1,2\}} \sum_{\substack{j=1 \\ j \neq i}}^{N+2} -f_t(\theta_t; x_i) log(f_s(\theta_s; x_j)) \tag{2}$$

### 3.2 Masked token strategy

**Random mask strategy.** Inspired of the MLM strategy for natural language pre-training, we apply the random mask strategy to self-supervised learning. Given a dataset $Q$ without manual annotations, and $\mathbf{e} = (e_1, ..., e_n)$ denote a image of $n$ tokens, where $i = 1, ..., n$. Let $\mathbf{m} = (m_1, ..., m_n)$ denote a binary vector of length $n$, where $m_i \in \{0, 1\}$, representing the mask over image. According to BERT [10], the $\mathbf{m}$ can be obtained with probability $p$ by Eq (3), and the $p$ is 0.15 by default.

$$m_i = \begin{cases} 1, & prob_i < p \\ 0, & \text{otherwise} \end{cases} \tag{3}$$

**Algorithm 1** Pseudo code of attention-guided mask strategy in a PyTorch-like style.

```
# l(): linear projection
# f_s: backbone + projection head
# f_t: backbone + projection head
# mask_embedding: learnable token
# p: mask probability

e_t = f_t.l(x) # linear projection
patch_attention, _ = f_t.Transformer(e_t)

importance = measure_importance(patch_attention) # acquire threshold

e_s = f_s.l(x) # linear projection
mask = M(e_s, patch_attention, importance) # generate mask
e_s_masked = (1-mask) * e_s + mask * mask_embedding
_, _ = f_s.Transformer(e_s_masked)

def M(embedding, patch_attention, importance):
    B, L, _ = embedding.shape
    mask_tokeep = zeros((B, L))
    mask_remove = bernoulli(ones((B, L)) * p)
    mask = where(importance > patch_attention, mask_tokeep, mask_remove)

    return mask
```

According to Eq (3), the tokens of crucial and nonessential regions have the same probability of being masked. As shown in Figure 2 (c), we observe that the random mask strategy may eliminate tokens of crucial regions that are responsible for recognizing objects, resulting in indistinguishable semantic features for input images. The random mask strategy is prone to mask crucial regions for images, and suppress the ability of network to recognize objects. It is not suitable to directly apply this strategy to self-supervised vision Transformers and the overall performance would deteriorate if the mask strategy is not properly modulated.

**Attention-guided mask strategy.** In this section, we propose our attention-guided mask strategy for dynamically controlling the fidelity of masked tokens and thereby decreasing the probability of masking crucial regions in self-supervised Transformer. Meanwhile, our strategy does not increase additional time consumption. Our algorithm is shown as Alg. 1.

Our framework consists of two networks, teacher network $f_t$ and student network $f_s$, with the same transformer architecture. Let $x$ denotes the input image. It is firstly projected to a sequence of $n$ 1-d tokens $\mathbf{e} = e_1, ..., e_n$, and then processed by several self-attention layers. Each self-attention layer [27] owns three groups of embeddings for one token, denoted as $Q_i$(query), $K_i$(key), $V_i$(value). The attention map is calculated as the correlation between the query embedding of class token $Q_{cls}$ and key embeddings of all other patches $K$. It is averaged for all heads as Eq (4). We output the attention map from the last layer in the teacher network to guide our strategy.

$$Attn = \frac{1}{H} \sum_{h=1}^{H} \text{Softmax}(Q_h^{cls} \cdot \frac{K_h^T}{\sqrt{d}}) \tag{4}$$

We sort the attention of different patches for each image in ascending order, and take the sorted attention value of $1/num$ of total tokens as the threshold $\tau$, where $num$ is the hyperparameter for selection. This means that the lowest $1/num$ of total tokens are selected as the masked candidates. The student model receives the importance of different patches and generates the mask $\mathbf{m}$ with probability $p$, according to the Bernoulli distribution as Eq (5). $prob_i$ refers to the probability of randomly generated.

$$m_i = \begin{cases} 1, & prob_i < p \text{ and } Attn_i < \tau \\ 0, & \text{otherwise} \end{cases} \tag{5}$$

We use $\mathbf{m} \odot \mathbf{e}$ to denote the final masked tokens as Eq (6). Follow the BERT [10], the masked regions are filled with a learnable mask embedding [MASK]. Our strategy can ensure the patches with the highest scores are always presented (in Figure 2).

$$(\mathbf{m} \odot \mathbf{e}) = \begin{cases} [\text{MASK}], & m_i = 1 \\ e_i, & m_i = 0 \end{cases} \tag{6}$$

The attention-guided mask strategy can benefit pre-training models in two ways:

1. The models utilize contextual information to understand the relationship of different patches, thus preserving the global semantic information of the image while paying more attention to the local details of the image.

2. Our strategy can avoid masking crucial regions while replacing nonessential regions with the learnable mask embedding, making the models focus on the crucial regions.

### 3.3 Masked self-supervised transformer

In MLM, $\overline{\mathbf{mask}}$ denotes the complementary set of $\mathbf{mask}$, that is, $\overline{\mathbf{mask}} = \mathbf{1} - \mathbf{mask}$. The loss function of MLM pre-training strategy over one data is shown as Eq (7), where $P(x_i|\theta, \mathbf{mask} \odot \mathbf{t})$ is the probability of the network correctly predicting $t_i$ given the masked token. $\mathbf{t}$ indicates the text tokens. That is, the network only restores the masked tokens.

$$l_{MLM}(\theta; t, m) = -logP(\overline{\mathbf{mask}} \odot \mathbf{t}|\theta, \mathbf{mask} \odot \mathbf{t}) = - \sum_{i:m_i=1} logP(t_i|\theta, \mathbf{mask} \odot \mathbf{t}) \tag{7}$$

There are a sub-sequence $M \subset [1, n]$ such that each index $i$ independently has probability $p$ of appearing in $M$, and the overall loss function for training the network is shown as Eq (8). $Q$ is the dataset in Eq (8). In pre-training, the MLM strategy minimizes the overall loss over pre-training dataset.

$$L_{MLM}(\theta) = \mathop{\mathbb{E}}_{\mathbf{t} \sim Q} \mathop{\mathbb{E}}_{M} l_{MLM}(\theta; \mathbf{t}, \mathbf{mask}). \tag{8}$$

However, MLM only predicts the masked tokens according to Eq (8). Different from original MLM, our method encourage the network reconstruct the original input images. We argue that a pixel-level restoration task can make the network avoid overfitting patch prediction, therefore enhancing the ability to capture the pixel-level information and recovering spatial structure from a finer grain. Since convolution neural networks (CNNs) have the ability of inductive biases, the restoration task adopts CNN as the decoder module, with convolution layers and up-sampling operations alternately stacked. To maximally mitigate the adversarial effect [37], the up-sampling operations are restricted to 2×. Hence, a total of 4 operations are needed for reaching the full resolution from $\frac{H}{16} \times \frac{W}{16}$. And the running mean and running variance of BN are only updated from the global views. The global image decoder consists of the Transformer and decoder. The restoration task is only performed on the student network $f_s(\cdot)$. For a decoder $g(\cdot)$ with parameters $\theta_g$, its loss function over a image $x \in R^{H \times W}$ and a mask $\mathbf{m} \in (0, 1)^n$ as Eq (9).

$$l_{restoring}(\theta_s, \theta_g; x, m) = \mathop{\mathbb{E}}_{H \times W} |x - g(\theta_g; f_s(\theta_s; x, m))| \tag{9}$$

The overall loss function for training the network is shown as Eq (10), and we only need the parameters $\theta_s$ of student network $f_s$.

$$L_{restoring}(\theta_s) = L_{restoring}(\theta_s, \theta_g) = \mathop{\mathbb{E}}_{\mathbf{x} \sim X} \mathop{\mathbb{E}}_{H \times W} |x - g(\theta_g; f_s(\theta_s; x, m))| \tag{10}$$

Therefore, the total loss is shown as Eq (11), and the MST minimizes the loss over ImageNet [9] dataset in pre-training.

$$L_{total}(\theta_s) = \lambda_1 * L_{CE}(\theta_s; x) + \lambda_2 * L_{restoring}(\theta_s; x) \tag{11}$$

Table 1: **Comparison of popular self-supervise learning methods on ImageNet.** Throughput (im/s) is calculated on a single NVIDIA V100 GPU with batch size 128. † adopts the linear probing of DINO.

| Method | Architecture | Parameters | epoch | im/s | Linear | k-NN |
|---|---|---|---|---|---|---|
| Supervised | | | 100 | 1237 | 76.5 | - |
| MoCov2 [6] | Res50[16] | 23 | 800 | 1237 | 71.1 | 61.9 |
| BYOL [13] | | | 1000 | 1237 | 74.4 | 64.8 |
| SwAV [1] | | | 800 | 1237 | 75.3 | 65.7 |
| Supervised | | | 300 | 1007 | 76.4 | - |
| SwAV [1] | | | 300 | 1007 | 67.1 | - |
| SimCLR [4] | | | 300 | 1007 | 69.0 | - |
| BYOL [13] | | | 300 | 1007 | 71.0 | - |
| MoCov3 [7] | | | 300 | 1007 | 72.5 | - |
| MOBY [30] | | | 300 | 1007 | 72.8 | - |
| BYOL [13] | | | 800 | 1007 | 71.4 | 66.6 |
| MoCov2 [6] | DeiT-S[26] | 21 | 800 | 1007 | 72.7 | 64.4 |
| SwAV [1] | | | 800 | 1007 | 73.5 | 66.3 |
| DINO [2] | | | 300 | 1007 | 75.2 | 72.8 |
| DINO† [2] | | | 300 | 1007 | 75.9 | 72.8 |
| DINO† [2] | | | 800 | 1007 | 77.0 | 74.5 |
| Ours † | | | 100 | 1007 | 75.0 | 72.1 |
| Ours | | | 300 | 1007 | **76.3** | **75.0** |
| Ours † | | | 300 | 1007 | **76.9** | **75.0** |
| Supervised | | | 300 | 755 | 81.2 | - |
| MoBY [30] | Swin-T[20] | 28 | 100 | 755 | 70.9 | 57.34 |
| Ours | | | 100 | 755 | **73.8** | **66.20** |

# 4 Experiments

Several experiments with MST are conducted in this section. We first train self-supervised models with different transformer architectures on ImageNet benchmark, and then examine their transfer capacity with downstream tasks like object detection and semantic segmentation. After that, ablation studies are introduced to elaborate on how our method could achieve state-of-the-art performance.

## 4.1 Pre-training settings

**Dataset and Models** Our method is validated on the popular ImageNet 1k dataset [9]. This dataset contains 1.28M images in the training set and 50K images in the validation set from 1000 classes. We only use the training set during the process of self-supervised learning. As to models, we choose the classical DeiT-S [26] and popular Swin-T [20] as representatives of all transformer-based architectures. After the backbone, a 3-layer MLP with hidden dimension 2048 is added as the projection head. When evaluating our pretrained model, we both use the k-NN algorithm and train a linear classification for 100 epochs as former works. Top-1 accuracy is reported.

**Training Configurations** Our model is optimized by AdamW [22] with learning rate $2 \times 10^{-3}$ and batch size 1024. Weight decay is set to be 0.04. We adopt learning rate warmup [12] in the first 10 epochs, and after warmup the learning rate follows a cosine decay schedule [21]. The model uses multi-crop similar to [1] and data augmentations similar to [13]. The setting of momentum, temperature coefficient, and weight decay follows [2]. The coefficient $\lambda_1$ of basic instance discrimination task is set as 1.0 while the restoration task $\lambda_2$ is set as 0.6.

## 4.2 Compared with other methods on ImageNet

We compare our method with other prevailing algorithms in Table 1. All these methods share the same backbone for fair comparison. Our 300-epoch model achieves 76.9% top-1 accuracy with linear probing. It outperforms previous best algorithm DINO by 1.7% at the same training epochs, and even approaches the performance of DINO with a much longer training schedule (77.0% with 800 epochs). It should be emphasized that our algorithm relieves the need of extreme long training time for self-supervised learning, and is able to obtain a decent result (75.0%) with only 100 epochs.

MST is general to be applied with any other transformer-based architectures. Here we use the popular Swin-T for an example. It has similar amount of parameters with DeiT-S. Using the same training epochs, MST outperforms MoBY by 1.8%, which is a self-supervised learning method designed delicately for Swin-T. Swin-T shares the same hyperparameters with DeiT-S, there it can still be improved by further tuning.

## 4.3 Object detection and instance segmentation

Since Swin-Transformer achieves state-of-the-art under supervised training, it is adopted as the backbone to validate the transfer ability of our method in the task of object detection and instance segmentation. We perform object detection experiments with MS COCO [18] dataset and Mask R-CNN detector [15] framework. MS COCO is a popular benchmark for object detection, with 118K images in training set and 5K images for validation. This dataset contains annotations for 80 classes. Box AP and mask AP are reported on the validation set. As to training settings, we follow the default 1x schedule with 12 epochs. The shorter edges of the input images are resized to be 800 and the longer edges are limited by 1333 pixels. AdamW optimizer is used, and all hyper-parameters follow the original paper.

In Table 2, we show the performance of the learned representation by different self-supervised methods and supervised training. For fair comparison, all these methods are pre-trained with 100 epochs. We observe that our method achieves the best results with 42.7% bbox mAP and 38.8% mask mAP. It outperforms the ImageNet supervised model by 1.2% and 0.5%, and MoBY results by 1.2% and 0.5% with the same epoch. The results indicate that MST not only performs well on image classification task, but also performs well on downstream dense prediction task. Therefore it has a strong transfer ability.

Table 2: Results of object detection and instance segmentation fine-tuned on MS COCO.

| Method | Backbone | Epoch | box AP | | | mask AP | | |
|---|---|---|---|---|---|---|---|---|
| | | | $AP^{bbox}$ | $AP^{bbox}_{50}$ | $AP^{bbox}_{75}$ | $AP^{mask}$ | $AP^{mask}_{50}$ | $AP^{mask}_{75}$ |
| Supervised | Swin-T [20] | 300 | 43.7 | 66.6 | 47.7 | 39.8 | 63.3 | 42.7 |
| | | 100 | 41.6 | 64.6 | 45.4 | 38.4 | 61.5 | 41.0 |
| MoBY [30] | | | 41.5 | 64.1 | 45.2 | 38.3 | 61.0 | 40.8 |
| DINO [2] | Swin-T [20] | 100 | 42.2 | 64.6 | 46.3 | 38.7 | 61.5 | 41.3 |
| Ours | | | **42.7** | **65.1** | **46.7** | **38.8** | **61.8** | **42.5** |

## 4.4 Semantic segmentation

SETR [37] provide a semantic segmentation framework for standard Vision Transformer. Hence, we adopt the SETR as the semantic segmentation strategy on Cityscapes [8]. Cityscapes contains 5000 images, with 19 object categories annotated in pixel level. There are 2975, 500, and 1525 images in training, validation, and testing set respectively. We follow the training config as original SETR. For fair comparison, we both use the 300-epoch pretrained model for DINO and our method.

As shown in Table 3, it illustrates the comparison of supervised method, DINO, and our method on this evaluation. Our method achieves the highest mIoU 74.7% and mAcc 82.35%. It outperforms both supervised results (+2.71% mIoU and +2.05% mAcc) and DINO pretrained results (+1.08% mIoU and +1.03% mAcc). Our model is also suitable to transfer for the semantic segmentation task.

Table 3: Results of semantic segmentation fine-tuned on Cityscapes.

| Method | Backbone | Pre-Epochs | Schedule | mIoU | mAcc | aAcc |
|---|---|---|---|---|---|---|
| Supervised | DeiT-S [26] | 300 | 40K | 71.33 | 80.30 | 94.99 |
| DINO [2] | DeiT-S [26] | 100 | 40K | 72.96 | 81.32 | 95.37 |
| Ours | | | | **74.04** | **82.35** | **95.42** |

Table 4: Linear probe results of different mask strategy (DeiT-S).

| Mask Strategy | Top-1 acc (%) |
|---|---|
| None | 73.1 |
| Random Mask | 63.2 |
| Attention-Guided | **73.7** |

Table 5: The setting of hyper-parameters for attention-based mask strategy.

| $num$ \ $p$ | 0.05 | 0.10 | 0.15 |
|---|---|---|---|
| 1 | 63.2 | 61.4 | 60.6 |
| 2 | 73.7 | 64.4 | 62.7 |
| 4 | 73.6 | 73.6 | 66.7 |
| 8 | 73.6 | **73.9** | 73.6 |

## 4.5 Ablation studies

In this section, we conduct some ablation studies to elaborate on the effectiveness of our method. All ablation experiments are conducted under 100-epoch setting. By default, only the *cls* token from the last layer is used to train the linear classifier.

### 4.5.1 Impact of different mask strategy

Table 4 shows the impact of different mask strategies. We train DeiT-S with random mask strategy[10], attention-guided mask strategy and no mask. For fair comparison, all methods mask with the same probability $p$. It can be observed that the performance of random mask strategy degrades. This strategy would probably suppress the ability to recognize the object in the images (from 73.1 to 63.2). Random mask strategy may destroy the tokens of crucial regions of original image which may be indispensable for recognizing object. The masked input may have incomplete or even misleading information. On the contrary, the performance of our attention-guided mask strategy has a steady improvement (from 73.1 to 73.7). Essential regions are mostly preserved, which could be a strong proof of our hypothesis.

### 4.5.2 Impact of different mask hyper-parameters

Table 5 validates the performance of different mask hyper-parameters under attention-guided mask strategy. We sort the attention map of different patches for each image in ascending order, and split the first $1/num$ patches as the masked candidates. Removing these candidates can force the network to learn local features from adjacent patches, therefore strengthening the capacity of modeling local context without destroying the semantics. These candidates are masked according to the probability $p$. Top-1 accuracy of linear evaluation on ImageNet is shown in Table 5. When $num$ is set to 8, any choice of $p$ can get a robust result, which suggests that the last $1/8$ patches are relatively safe to be mask candidates.

## 4.6 Impact of w/o BN

Former work [2] found that the performance will be better if dropping BN in the projection head. We argue that the degradation is not caused by BN. As shown in Table 6, normal BN downgrades the performance of baseline model, while the update rule introduced in Section 3.3 helps improve top-1 accuracy slightly. This may be due to the need to keep consistent structure with the global image Decoder since the image Decoder consists of Conv-BN-ReLu.

Table 6: Impact of Batch Normalization in projection head.

| | w/o BN | w/ BN |
|---|---|---|
| Baseline | 72.4 | 71.6 |
| Ours | 73.1 | **73.9** |

# 5 Conclusion

In this paper, we investigate the two problems of current visual self-supervised learning, namely lack of local information extraction and loss of spatial information. To overcome the above problems, we propose a new self-supervised learning method based on transformer called MST. The proposed MST exploits an attention-guided mask strategy to capture the local relationships between patches while also preserving the global semantic information. It is noted that the attention-guided mask strategy is based on the multi-head self-attention map extracted from the teacher model and does not cause extra computation cost. In addition, a global image decoder is further used under the attention-guided mask strategy to recover the spatial information of the image, which is vital for dense prediction tasks. The proposed method shows good versatility and scalability in multiple downstream visual tasks.

## Broader Impact

The MST is to provide pre-trained models with better feature extraction capabilities for popular computer vision tasks. Therefore, the potential positive societal impact of our method:

- Improved road safety in autonomous driving by detecting pedestrians.
- Safe human robot collaboration in factories with robots taking on risk prone jobs thus saving lives.
- Assistive robots for elderly care.

Meanwhile, the potential negative societal impact:

- Large scale drone surveillance.
- Face recognition leads to privacy exposure.

## Acknowledgments and Disclosure of Funding

This work was supported by National Natural Science Foundation of China under Grants No.61772527, No.61976210, No.62002357, No.61876086, No.61806200, No.62002356, No.62006230, and No.62076235.

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
