# OpenReview forum: "MST: Masked Self-Supervised Transformer for Visual Representation"
_NeurIPS.cc/2021/Conference — NeurIPS 2021 Poster_

### Official Review · Reviewer_hFAf · 2021-07-16

**Rating:** 6
**Confidence:** 4

**Summary:**

The paper proposes to add an image reconstruction task to the existing instance discrimination task, as the pre-text tasks for self-supervised learning. Specifically, the self-attention map from the teacher model is used to guide the random mask process, that the attended foreground regions will not be masked. A decoder is used to restore the original image from the masked image.

**Limitations And Societal Impact:**

The authors didn't provide a statement of broader impact.

**Main Review:**

The overall idea of combining image reconstruction and instance discrimination as pre-text tasks makes sense.
However, the paper is poorly written. Both the ablations and analysis are insufficient to demonstrate the effectiveness of the proposed method.

- The authors are suggested to provide the experiments of only removing the reconstruction loss. Because the gains could be from the data augmentation itself.
- More ablation study is needed. For example, random mask strategy with different sampling ratios,  the impact of loss weight $\lambda_2$.
- The authors are suggested to demonstrate their method by first reporting the results of the baseline and then adding each small component of the proposed method.
- L181, what's 'global crops'.
- The authors should pay attention to the notations. For example, what's 'Q' in Eq.(8), 't' on L167, 'num' on L152, 'd' on Eq.(4), 'prob_j' in Eq.(3). Although I can get the meaning by guess or by experience, they should be clearly explained.


Other minors:
- L20-L21, it's meaningless to report a single number on downstream detection and segmentation tasks. Reporting the gains from the reconstruction task should make more sense.
- L225, COCO has 80 categories, not 81.
- Many typos and grammar errors, e.g., L25, L57, L114, L145, L166.

## After Rebuttal
The rebuttal and discussion have addressed most of my concerns. Thus I raise my rating from 4 to 6.
The authors are suggested to add all these ablations and discussion, and further revise the paper, especially the presentation.

**Time Spent Reviewing:**

6 hours

---

> ### Author Response · Authors · 2021-08-10
> **Response**
>
> Thank you for the positive comments and constructive feedback. Below are our responses to specific comments.
>
> **Question 1**: The authors are suggested to provide the experiments of only removing the reconstruction loss. Because the gains could be from the data augmentation itself.
>
> **Response**: We conduct the experiment by removing the reconstruction loss, the linear evaluation result is 72.0% (+0.4%), which is only slightly better than the baseline (71.6%, in Table 6).
> Meanwhile, our method (73.9%) outperforms the result by 1.9%. Hence, we argue that the gains are not mainly from the data augmentation.
>
> **Question 2**: More ablation study is needed. For example, random mask strategy with different sampling ratios, the impact of loss weight.
>
> **Response**: In the first line of Table 5, we already show the results of the random mask strategy with different sampling ratios. We also have tried a small p (0.01) with random masking, the result is 71.1%. When the p is smaller, the performance will be better. The result is best (72.6%, contrastive loss + restore loss) when p is set to 0. Empirically, we set the restoration coefficient \lambda_2 to 0.6, which makes the contrastive loss and the restoration loss roughly equally weighted. We have also tried several different settings of \lambda_2 (e.g., 0.2, 0.4, 0.6, 0.8), the result is 73.7%, 73.5%, 73.9% and 73.6% respectively. It can be observed that the results are also insensitive to \lambda_2, and the best performance is achieved when the two losses are equally weighted.
>
> **Question 3**: The authors are suggested to demonstrate their method by first reporting the results of the baseline and then adding each small component of the proposed method.
>
> **Response**: As mentioned in Question 1 and Question 2, we acquire the result of the baseline is 71.6%, the baseline + mask is 72.0%, and baseline + mask + restore loss is 73.9%. We will provide a clearer version in the revised manuscript.
>
> **Question 4**: L181, what's 'global crops'.
>
> **Response**: In lines 118-120, referring to other advanced self-supervised methods (e.g., DINO, MOCOv3), we also adopt multi-crop data augmentation strategy to generate two global crops and N local crops. Following the setting of DINO, the global crops are cropped from the 0.4-1.0 size of the original image while local crops are cropped from the 0.05-0.4 size of the original image.
>
> **Question 5**: The authors should pay attention to the notations. For example, what's 'Q' in Eq.(8), 't' on L167, 'num' on L152, 'd' on Eq.(4), 'prob_j' in Eq.(3). Although I can get the meaning by guess or by experience, they should be clearly explained.
>
> **Response**:
>
> * As shown in the below equation, Q is the dataset in Eq(8). 't' on L167 indicates the text tokens in MLM.
>
> $L_{MLM}(\theta) =
>  \mathop{\mathbb{E}}\limits_{\mathbf{t}\sim Q} \mathop{\mathbb{E}}\limits_{M} l_{MLM}(\theta;\mathbf{t}, \mathbf{mask})$
>
>
>
> * 'num' on L152 is the masked candidates' hyperparameter for selection.
>
> * As shown in the below equation, 'd' on Eq(4) is the dimension of per head in Transformer.
>
> $Attn =
>  \frac{1}{H}\sum_{h=1}^{H} Softmax(Q_h^{cls}\cdot \frac{K_h^T}{\sqrt{d}})$
>
> * As shown in the below equation, 'prob_j' in Eq.(3) refers to the probability of randomly generated.
>
> $\begin{equation}
>  m_i =
> \begin{cases}
> 1, &  prob_i < p  \\\\
> 0, &  otherwise
> \end{cases}
> \end{equation}$
>
> All of these notations’ minors will be fixed in the revised manuscript.
>
> **Other minors**: We thank your careful comments, and kindly be informed all of the minors and typos will be corrected in the revised manuscript.
>
> **Summary**: In this paper, we point out two problems of current visual self-supervised learning: (a) the instance discrimination methods use the global features, but lack of local information extraction; (b) the masked language modeling（MLM) is prone to mask the tokens of crucial regions for images but lost spatial (global) information (the MLM masks some tokens randomly and just predicts the original masked tokens). To overcome the above problems, in this paper, the instance discrimination method is imported from the theory of MLM to capture the local relationships between patches while also understanding the global features.  Specifically, our contributions mainly include: (a) We propose an attention-guided mask strategy to enhance the understanding of local context semantics in pre-training without damaging the crucial structure. As far as we know, this is the first work that deeply studies the MLM strategy in vision pre-training; (b) we propose to exploit a global decoder to further recover the spatial information, which greatly improves the versatility and scalability of the pre-training model.
>
> The main problems focus on the effectiveness of mask strategy and each small component of the proposed method. From the responses of Question 2 and Question 3, it can be observed that the proposed method can enhance the performance of pre-trained model and the gains are not mainly from the masking augmentation strategy or restore loss alone. We will also try to make the details clearer and report the results by adding each small component of the proposed method. In conclusion, the ablations and analysis are sufficient to demonstrate the effectiveness of the proposed method.

---

> > ### Comment · Reviewer_hFAf · 2021-08-25
> > **Thanks for the rebuttal**
> >
> > Thank you for the rebuttal.
> > The rebuttal provides new experiments which address most of my concerns, e.g., the net gains of restoration loss and the impact of the loss weight.
> > I'm happy to see that the authors provide more ablations, e.g., MLM task alone, and reconstruct full input image vs. masked tokens. I agree with Reviewer 2YYa that the ablations with some negative results should be further analyzed as they should make this paper more solid.
> >
> > I only have one minor question. I'm still confused about the result of not using the mask strategy.
> > > The result is best (72.6%, contrastive loss + restore loss) when p is set to 0.
> > While Table 4 reports 73.1% for the 'None' mask strategy.
> > To me these two should be the same experiments, right?

---

> > > ### Author Response · Authors · 2021-08-26
> > > **Response**
> > >
> > > In fact, there is a slight difference between the two results. The ‘None’ mask strategy without BN is 72.6% while the ‘None’ mask strategy with BN is 73.1% in Table4 (the extra BN ablation experiments are shown in Section 4.6 and Table 6). For consistent comparison, we show the results with BN in our paper (from Table 1 to Table 5). For smaller p (e.g. 0.01), we have early result without BN, thus we also report the result without BN in the rebuttal (that’s 72.6%). However, regardless of BN, these results confirm our conclusion “The result is best when p is set to 0”.
> > >
> > > We appreciate the careful comments of the reviewers. We will add these details and analysis to the revised manuscript to make the paper more solid.

---

> > > > ### Comment · Reviewer_hFAf · 2021-08-27
> > > > **Thanks for the rebuttal**
> > > >
> > > > Now it's clear. This addresses my concern.
> > > >
> > > > I would like to raise my rating considering the provided more ablations and analysis.
> > > > Please add all these ablations and further revise the paper, especially the presentation.

---

> > > > > ### Author Response · Authors · 2021-08-27
> > > > > **Thanks for your careful comments**
> > > > >
> > > > > Thanks, we will add all these ablations and further revise the paper.

---

### Official Review · Reviewer_2YYa · 2021-07-16

**Rating:** 7
**Confidence:** 3

**Summary:**

This paper presents a new self-supervised learning method based on a combination of mean-teacher distillation (in this case DINO) and masked patch modelling (like masked language modelling in NLP). They use this to train transformer based vision models and show downstream results for ImageNet classification using linear probing and knn eval, MS COCO object detection and instance segmentation using Mask-RCNN framework and Cityscapes semantic segmentation using SERT framework.

Their key technical contributions are (a) the overall training framework combining DINO with masked patch modelling. The latter requires a CNN decoder and reconstructs the original global image and not just the masked patches and (b) a smart mask generation scheme based on the attention map of the CLS token which turns out to be crucial for good results. They use the attention maps corresponding to the CLS token to select patches that are not important. The idea is that if important regions of the image are masked out then the model will end up modelling only local statistics.

At a deeper level, their method is trying to retain spatial structure in the latent representation while capturing global image properties. If I understood correctly, the spatial structure comes from masked patch modelling and global image properties come from DINO.

**Limitations And Societal Impact:**

Limitations are not discussed. What would be something the community should address moving forward?

Societal impact is not discussed. The authors state that "This work dose not present any foreseeable societal consequence.". The method improves object detection and instance segmentation performance. These datasets prominently include the "person" category. Some suggestions are below.

Potential positive societal impact:
* Improved road safety in autonomous driving by detecting pedestrians.
* Safe human robot collaboration in factories with robots taking on risk prone jobs thus saving lives.
* Assistive robots for elderly care.

Potential negative societal impact:
* Large scale drone surveillance.

The societal impact of anything in computer vision is massive! I agree that the above areas are not a direct outcome of the proposed method but are a step in that direction.

There are no error bars. The authors say error bars are in the appendix. But they are missing. Training for 100 epochs makes it super difficult to repeat the experiment 3-5 times. If possible, maybe just run the main proposed setting 3 times at 100 epoch schedule and do knn eval on imagenet. This way we can get some idea of the variance.

**Main Review:**

Both technical contributions listed above are novel to the best of my knowledge. The method is of significance to the community as self-supervised learning now outperforms supervised pre-training and could potentially help the community save on data annotation costs. It is also significant in that a lot of prior self-supervised learning literature learns global image representations which may not adequately preserve spatial structure. The proposed method addresses this issue.

The performance on established benchmarks is strong. The explanation is mostly clear. Some areas for improving the paper are discussed below.

Citations
------------
Please cite
* [1] Self-Supervised Representation Learning from Flow Equivariance. Yuwen Xiong, Mengye Ren, Wenyuan Zeng, Raquel Urtasun, Tech report, arXiv, Jan. 2021 which like DenseCL and PixPro also learns dense image representations for downstream detection and segmentation tasks.
* [2] Unsupervised Learning of Dense Visual Representations. Pedro O. Pinheiro, Amjad Almahairi, Ryan Y. Benmalek,
Florian Golemo, Aaron Courville, NeurIPS 2020 is also such a method.
* [3] https://papers.nips.cc/paper/2020/file/c1502ae5a4d514baec129f72948c266e-Paper.pdf also focus on downstream segmentation and identify the same issue that MoCo learns only global image representations.
* In line 179, what does adversarial effect refer to? A citation here might help clarify what the authors mean.
* Numbers for prior methods in the experiments section were in some cases taken from prior papers. Making the source of each number explicit in the table caption will make it easier for the reader to dig out the technical details for each entry. Also especially clarify if any of these numbers are based on you running the opensource code from prior work.
* In line 201, does "former works" refer to DINO?
* Vision Transformers are a new phenomenon. Unlike ResNet and VGG most readers would not know what DeIT-S and Swin-T are. A 3-5 line summary of these models will be of great value in the related works section.

Important missing ablations
---------------------------------------
* What is the performance of just L_restoring? Does it have to be combined with DINO for this to work? Concurrent work on MLM for Vision Transformer (https://arxiv.org/pdf/2106.08254.pdf) is able to work purely using an MLM objective. Answering this question in an ablation experiment will be super useful to the community.
* In line 174, the authors mention that unlike original MLM where the loss only looks at masked tokens, they reconstruct the full input image. Please include an ablation to support this design choice. Also clarify how multi scale cropping effects the MLM part of the method. Is the MLM part also exposed to lower resolution crops?
* In line 178, the authors argue how inductive biases in CNNs inspire them to use CNNs to decode the image. One could alternatively just decode using a projection into 16x16x3 patches and concatenate them together, or decoder into a lower resolution image. Basically how crucial is it to have the CNN inductive biases in this part of the training framework?
* Masking is done after the linear projection. I found this only when looking carefully at figure 1. Is this important?

Differences with BERT
------------------------------
* BERT randomly replaces the masked positions with the learned [mask] token or with other words. The image equivalent would be to replace chosen patches with either the [mask] token or other patches. However, the proposed method only replaces chosen patches with [mask]. Did the usual BERT approach not work? Sorry I am not familiar with NLP literature, so maybe this is no longer the current practice in NLP either.
* Do NLP methods also need to be some attention guided masking? If not, then why do those methods not suffer from masking out important words during training?

Minor issues
-----------------
* Table 6 caption: Impact of batch normalization "in projection head".
* Line 283, global image decoder is further used "under" the attention-guided mask ...
* Line 167, 't' is a new variable. Did you mean 'e'?
* Line 196, 50k images not 5k.
* Line 183, what is $H^n$?

**Time Spent Reviewing:**

6

---

> ### Author Response · Authors · 2021-08-10
> **Response**
>
> Thank you for the positive comments and constructive feedback. Below are our responses to specific comments.
> # Citations
>
> **Question 1**: In line 179, what does adversarial effect refer to? A citation here might help clarify what the authors mean.
>
> **Response**: Directly up-sampling more than 2x is easy to produce unstable results. Here, we will cite some citations (for example, [1]) in the revised manuscript.
>
> **Question 2**: In line 201, does "former works" refer to DINO?
>
> **Response**: Former works refer to self-supervised vision transformer methods, like MoCov3 [2], DINO [3], and MOBY [4].
>
> **Question 3**: Other citations problems.
>
> **Response**: We will cite these papers and make the details clearer in the revised manuscript.
>
> # Important missing ablations
>
> **Question 1**: What is the performance of just L_restoring? Does it have to be combined with DINO for this to work? Concurrent work on MLM for Vision Transformer is able to work purely using an MLM objective. Answering this question in an ablation experiment will be super useful to the community.
>
> **Response**: In the former work ViT [5], Section 4.6 of ViT paper shows the supervised method outperforms the MLM by 4% with ViT-B despite using JFT-300M dataset. Therefore, this conclusion enlightens us that the pure L_restoring can not achieve better performance. The concurrent work BEiT also exploits the MLM as the pretext task and further predicts the reconstructed image. However, it is a two-stage rather than an end-to-end method, and just evaluates the performance of the pre-trained model by finetuning instead of the popular linear probing or k-NN. Hence, it’s hard to confirm its real performance. We will further investigate the finetuning results of our method as a comparison.
>
> **Question 2**: In line 174, the authors mention that unlike original MLM where the loss only looks at masked tokens, they reconstruct the full input image. Please include an ablation to support this design choice. Also clarify how multi scale cropping effects the MLM part of the method. Is the MLM part also exposed to lower resolution crops?
>
> **Response**: We applied the original MLM to our framework and found the result is lower than single contrastive learning (baseline). Hence, we reconstruct full input image instead of masked tokens and the pixel-level restoration task can make the network avoid overfitting patch prediction, therefore enhancing the ability to capture the pixel-level information and recovering spatial structure. Following the setting of DINO, we completely adopt the multi-crop strategy and its hyperparameters. For decreasing the calculation of pre-trained stage, we only apply the MLM to the global crops. In fact, the MLM can be exposed to lower resolution crops. We will further investigate the lower resolution version in future work.
>
> **Question 3**: In line 178, the authors argue how inductive biases in CNNs inspire them to use CNNs to decode the image. One could alternatively just decode using a projection into 16x16x3 patches and concatenate them together, or decoder into a lower resolution image. Basically how crucial is it to have the CNN inductive biases in this part of the training framework?
>
> **Response**: In the paper of ViT, the authors think that Transformer yields modest accuracies of a few percentage points below ResNets of comparable size may be due to lack some of the inductive biases inherent to CNNs when trained on ImageNet. Moreover, the decoder is only used to restore the image, and what kind of network structure to adopt is not the focus of self-supervised method. Hence, we only attempt to adopt CNN as a decoder in the pre-trained stage. We will attempt to decode by a linear projection or a lower resolution image.
>
> **Question 4**: Masking is done after the linear projection. I found this only when looking carefully at figure 1. Is this important?
>
> **Response**: The goal of linear projection is to map the image patches into tokens/embeddings. Following the setting of MLM, the masking (token) strategy should be done after linear projection. In Figure 2, the caption also describes that the masked strategy is to mask token. We will further make this detail clear in the revised manuscript.
>
> # Differences with BERT
>
> **Question 1**: BERT randomly replaces the masked positions with the learned [mask] token or with other words. The image equivalent would be to replace chosen patches with either the [mask] token or other patches. However, the proposed method only replaces chosen patches with [mask]. Did the usual BERT approach not work? Sorry I am not familiar with NLP literature, so maybe this is no longer the current practice in NLP either.
>
> **Response**: Each token represents a word or sub-word (corresponding to a category) in NLP, so the BERT mainly uses cross-entropy loss as the pretext loss and predicts the masked token from the vocabulary dictionary. Different from the classification loss in NLP, here we use the reconstruction loss to recover the image patch that is masked. It is not appropriate to replace the masked patches with other patches, which may mislead the model.  Therefore, we use the unified [mask] to represent the masked patches.
>
> **Question 2**: Do NLP methods also need to be some attention guided masking? If not, then why do those methods not suffer from masking out important words during training?
>
> **Response**: Visual self-supervised pre-training always adopts the contrastive loss to learn the similar feature representations of two image views. Hence, the proposed attention-guided mask strategy aims to avoid the missing of crucial features between two image views caused by the traditional MLM in NLP. The differences between visual and NLP pre-training tasks lead to subtle variations in masking strategy.
>
> # Minor issues
>
> Thanks for your suggestions. All the minors will be fixed in the revised manuscript.
>
> # Limitations And Societal Impact
>
> **Question 1**: Societal impact is not discussed. The authors state that "This work dose not present any foreseeable societal consequence.". The method improves object detection and instance segmentation performance. These datasets prominently include the "person" category. Some suggestions are below.
>
> **Response**: Thanks for your suggestion. We will discuss the positive and negative societal impacts in the revised manuscript.
>
> **Question 2**: There are no error bars. The authors say error bars are in the appendix. But they are missing. Training for 100 epochs makes it super difficult to repeat the experiment 3-5 times. If possible, maybe just run the main proposed setting 3 times at 100 epoch schedule and do knn eval on imagenet. This way we can get some idea of the variance.
>
> **Response**: In Appendix, Section A shows the same random seed is set for a fair comparison. Also, we report the average result after running multiple experiments.
>
> [1] Zheng, S., Lu, J., Zhao, H., Zhu, X., Luo, Z., Wang, Y., Fu, Y., Feng, J., Xiang, T., Torr, P.H.S., Zhang, L.: Rethinking semantic segmentation from a sequence-to-sequence perspective with transformers. arXiv preprint arXiv:2012.15840 (2021)
>
> [2]  Chen, X., Xie, S., He, K.: An empirical study of training self-supervised vision transformers. arXiv preprint arXiv:2104.02057 (2021)
>
> [3] Caron, M., Touvron, H., Misra, I., Jégou, H., Mairal, J., Bojanowski, P., Joulin, A.: Emerging properties in self-supervised vision transformers. arXiv: Computer Vision and Pattern Recognition (2021)
>
> [4] Xie,Z.,Lin,Y.,Yao,Z.,Zhang,Z.,Dai,Q.,Cao,Y.,Hu,H.: Self-supervised learning with swin transformers. arXiv preprint arXiv:2105.04553 (2021)
>
> [5] Dosovitskiy A, Beyer L, Kolesnikov A, et al. An image is worth 16x16 words: Transformers for image recognition at scale. arXiv preprint arXiv:2010.11929 (2020)

---

> > ### Comment · Reviewer_2YYa · 2021-08-16
> > **Ablations: Masking strategy and pure MLM baseline**
> >
> > Thank you for addressing my questions. I would like to follow-up on two points about which I am not entirely satisfied with the author's response.
> >
> > In MLM pre-text task, when replacing all masked tokens with [mask] the network is encouraged to pay special attention to [mask]. This is further aggravated when the loss is also only on these tokens. This hinders test time generalization because [mask] token is not present in test sequences. BERT overcomes this problem using smart masking - replace some masked tokens with other tokens and leave some unchanged in addition to replacing some with [mask]. The current submission proposes an alternative solutions: replace all masked tokens with [mask] and reconstruct all tokens in the target. The BERT solution was used by ViT[5] in section 4.6. Which of these is better in the computer vision setting? How big is the performance delta between them? Do both of these perform better than training from scratch? The authors argue that BERT style mask replacement is not suitable for vision because "not appropriate to replace the masked patches with other patches, which may mislead the model.". This requires empirical justification.
> >
> > Answering the above question in the "random mask" setting would also provide the much needed comparison between pure DINO and pure MLM. This is something the ViT paper said was left for future work. The ViT numbers in section 4.6 of their paper are not directly comparable with the numbers in this submission because of different loss, different dataset, different mask replacement strategy.
> >
> > These ablations will provide some negative results which will further strengthen the technical contributions of this paper, highlighting not only why attention guided masking is important but also how one needs to be careful not to mislead the model by replacing tokens with other tokens instead of [mask].

---

> > > ### Author Response · Authors · 2021-08-20
> > > **Response**
> > >
> > > We conduct the experiment by using pure MLM with DeiT-S under 100 epochs, the result is about 40% with the same experimental configuration. Then we further adjust its learning rate and other hyperparameters, the best result is only 61%, which is far lower than that of the DINO by 10.6% (71.6% in Table 6) and also lower than the vanilla supervised result by 7.7% (68.7%, the ViT paper provides the vanilla results without data augmentation, but does not show the result of ViT-S (DeiT-S), we acquire the result from the Table 2 of paper [1]). It shows the pure MLM method may be not suitable for computer vision tasks. Moreover, We experiment with the contrastive loss + BERT solution (that’s DINO+pure MLM), the linear result is 71.9%. Our method outperforms its result by 2.0% (73.9%). The result proves our method is better than the original MLM method. Meanwhile, we further conduct the experiment by only replacing the [mask] token with the strategy of pure MLM for our method, the linear result is 73.5%, which also behinds our result. These results fully demonstrate the better setting of MLM for computer vision and further highlight the technical contributions of our paper.
> > >
> > > [1] Heo, Byeongho and Yun, Sangdoo and Han, Dongyoon and Chun, Sanghyuk and Choe, Junsuk and Oh, Seong Joon. "Rethinking spatial dimensions of vision transformers." In ICCV, 2021.

---

> > > ### Author Response · Authors · 2021-08-20
> > > **Response**
> > >
> > > We conduct the experiment by using pure MLM with DeiT-S under 100 epochs, the result is about 40% with the same experimental configuration. Then we further adjust its learning rate and other hyperparameters, the best result is only 61%, which is far lower than that of the DINO by 10.6% (71.6% in Table 6) and also lower than the vanilla supervised result by 7.7% (68.7%, the ViT paper provides the vanilla results without data augmentation, but does not show the result of ViT-S (DeiT-S), we acquire the result from the Table 2 of paper [1]). It shows the pure MLM method may be not suitable for computer vision tasks. Moreover, We experiment with the contrastive loss + BERT solution (that’s DINO+pure MLM), the linear result is 71.9%. Our method outperforms its result by 2.0% (73.9%). The result proves our method is better than the original MLM method. Meanwhile, we further conduct the experiment by only replacing the [mask] token with the strategy of pure MLM for our method, the linear result is 73.5%, which also behinds our result. These results fully demonstrate the better setting of MLM for computer vision and further highlight the technical contributions of our paper.
> > >
> > > [1] Heo, Byeongho and Yun, Sangdoo and Han, Dongyoon and Chun, Sanghyuk and Choe, Junsuk and Oh, Seong Joon. "Rethinking spatial dimensions of vision transformers." In ICCV, 2021.

---

> > > > ### Comment · Reviewer_2YYa · 2021-08-24
> > > > **Ablation: Masking strategy and pure MLM baseline**
> > > >
> > > > Thank you for running these experiments.
> > > >
> > > > In the 100 epoch setting:
> > > >
> > > > ```
> > > > BERT style masking + reconstructing masked tokens only:        61%
> > > > BERT style masking + reconstructing masked tokens only + DINO: 71.9%
> > > > BERT style masking + reconstructing all tokens + DINO:         73.5%
> > > > Proposed masking   + reconstructing all tokens + DINO:         73.9%
> > > > ```

---

> > ### Comment · Reviewer_2YYa · 2021-08-24
> > **Error bars**
> >
> > "Also, we report the average result after running multiple experiments." If you still have those runs, please also report the standard error alongside the average. How many seeds were used?

---

> > > ### Author Response · Authors · 2021-08-26
> > > **Response**
> > >
> > > Thanks for your careful comment. We used 4 different seeds for running these experiments. For 100 epochs, the stanard error of linear probing is 0.2236%. For 300 epochs, the stanard error of linear probing is 0.1581%.

---

> > > > ### Author Response · Authors · 2021-08-26
> > > > **Data**
> > > >
> > > > The 100 epochs results are 74.2, 73.8, 74.0, and 73.6.Meanwhile, the 300 epochs results are 76.8, 77.1, 76.7, and 77.0.

---

> > > > > ### Comment · Reviewer_2YYa · 2021-08-26
> > > > > **Error bars**
> > > > >
> > > > > Thanks! Please mention the standard error (or individual results) and the number of seeds in the appendix.

---

> > > > > > ### Author Response · Authors · 2021-08-27
> > > > > > **Response**
> > > > > >
> > > > > > Thanks, we will add all these details to the revised appendix.

---

### Official Review · Reviewer_KqLC · 2021-07-18

**Rating:** 6
**Confidence:** 4

**Summary:**

This submission combine the task of mask language modeling (MLM) and a contrastive learning task for self-supervised visual representation learning. The authors claim that due to he preserve of spatial information of an image, the proposed approach is more friendly to the downstream dense prediction tasks. The effectiveness of the approach is verified on ImageNet-1K linear evaluation task, and downstream tasks of COCO object detection and ADE20K semantic segmentation.

**Ethics Review Area:**

["I don’t know"]

**Limitations And Societal Impact:**

Please see the questions above.

**Main Review:**

Originality

-- While the MLM alone shows inferior performance (iGPT and in ViT), this submission tries MLM as a complementary pretext task to the conventional contrastive method, and demonstrates superior performance than contrastive learning alone. It also proposes a new masking strategy according to the importance of a token. The originality is acceptable for NeurIPS but not outstanding.

Quality

-- The quality is borderline. The combination approach makes sense. But the effects of specific designs are not well ablated, and it is thus unclear to the readers what the real effective components are:
1) what is the performance by contrastive learning alone? Masking can be also regarded as a kind of data augmentation. Does it help?
2) what is the performance by mask language modeling alone?
3) can the two pretext tasks really benefit each other？ what is the effect if \lambda_2 varies?
4) Is random masking well tuned? seems it may need very small p, but the results at smaller p such as 0.01 are not provided. Does the poor performance by random masking come from poor contrastive learning or poor mask language modeling.
5) what would it perform with longer training, e.g. 500 or 800 epochs?
6) why there are no finetuning results on COCO and ADE using 300-ep pretrained models?
7) What is the speed compared to contrastive approaches? It seems that the masking strategy requires a pre-compute of the teacher network, and the two branches must be conduct in sequential. Does this sequential computation affect running speed?

Clarity

--Overall, the method is presented clear. But there lack a lot of details and hyper-parameters: are the two image views the same cropping/resizing augmentation? it seems to be different as in Line 205, but if the two views are different, how do you estimate \tau by the teacher network as their patches are not the same?

Significance

--Satisfied if the components are solidly verified.




 ablation is not

**Time Spent Reviewing:**

4 hours

---

> ### Author Response · Authors · 2021-08-10
> **Response**
>
> Thanks for your valuable suggestions and comments.
>
> **Question 1**: What is the performance by contrastive learning alone? Masking can be also regarded as a kind of data augmentation. Does it help?
>
> **Response**: In Table 6, it can be observed that the result of contrastive learning alone (projection head with BN) is 71.6%. We also conduct the experiment by adding the masking augmentation, the linear evaluation result is 72.0%, which is only slightly better than the baseline (71.6%). Meanwhile, our method (73.9%) outperforms the result by 1.9%. Hence, we argue that the gains are not mainly from the masking augmentation.
>
> **Question 2**: What is the performance by mask language modeling alone?
>
> **Response**: The former work ViT [1] has already shown inferior results while using MLM alone, the Section 4.6 of ViT paper shows the supervised method outperforms the self-supervised MLM by 4% with ViT-B despite using the JFT-300M dataset. Therefore, this conclusion enlightens us that the pure MLM is prone to mask the tokens of crucial regions for images and is hard to achieve better performance.
>
> **Question 3**: Can the two pretext tasks really benefit each other? what is the effect if \lambda_2 varies?
>
> **Response**: As mentioned the Question 1 and Question 2, we observe that our result is better than any single pretext task. Thus the two pretext tasks are benefited each other. Empirically, we set the restoration coefficient \lambda_2 to 0.6, which makes the contrastive loss and the restoration loss roughly equally weighted. We have also tried several different settings of \lambda_2 (e.g., 0.2, 0.4, 0.6, 0.8), the result is 73.7%, 73.5%, 73.9% and 73.6% respectively. It can be observed that the results are also insensitive to \lambda_2, and the best performance is achieved when the two losses are equally weighted.
>
> **Question 4**: Is random masking well tuned? seems it may need very small p, but the results at smaller p such as 0.01 are not provided. Does the poor performance by random masking come from poor contrastive learning or poor mask language modeling.
>
> **Response**: In fact, we also have tried a small p (0.01) with random masking, the result is 71.1%. When the p is smaller, the performance will be better. The result is best (72.6%, contrastive loss + restore loss) when p is set to 0, showing that the random masking may destroy the crucial structure for self-supervised learning. However, the performance of the pre-trained model would improve when the random mask strategy is replaced with the proposed attention-guided mask strategy (73.9%). Hence, we argue that the poor performance of the random mask strategy is not related to contrastive learning or MLM.
>
> **Question 5**: What would it perform with longer training, e.g. 500 or 800 epochs?
>
> **Response**: Due to the tremendous time-consuming pre-training with more epochs, here we only train at most 300 epochs. As shown in Table 1, our 300-epoch model outperforms the 800-epoch models of most other state-of-the-art methods and achieves a similar result to the DINO of 800-epoch (76.9% vs 77.0%). These phenomena show that our method has the advantage of fast convergence. We will further investigate longer pre-training with 800 epochs.
>
> **Question 6**: Why there are no finetuning results on COCO and ADE using 300-ep pretrained models?
>
> **Response**: In Table 3, it can be observed that the performance of the 100-epoch pre-trained model already outperforms the 300-epoch supervised model for semantic segmentation. It is sufficient to demonstrate the effectiveness of our method, thus we do not evaluate the 300-epoch model. In here, we also experiment with the performance of the 300-epoch model and find its result outperforms the 100-epoch by 0.4%. For object detection, there is no popular method based on the vanilla vision Transformer (DeiT, ViT). Therefore, we temporarily choose Swin-Transformer as the backbone for pre-training. Here we only train at most 100 epochs due to the tremendous time-consuming pre-training with more epochs. We will provide the longer pre-training results in the revised manuscript.
>
> **Question 7**: What is the speed compared to contrastive approaches? It seems that the masking strategy requires a pre-compute of the teacher network, and the two branches must be conduct in sequential. Does this sequential computation affect running speed?
>
> **Response**: The two branches（the teacher and the student network）are also conducted in sequential for the existing contrastive learning methods due to python itself is executed sequentially. Hence, our method hardly affects the running speed since it only needs to compute the self-attention map from the last layer in the teacher feedforward. Section G in the Appendix shows the pseudo code of our method.
>
> **Question 8**: Are the two image views the same cropping/resizing augmentation? it seems to be different as in Line 205, but if the two views are different, how do you estimate \tau by the teacher network as their patches are not the same?
>
> **Response**: In the data augmentation stage, we **fully follow the settings of DINO** (Section B in Appendix), including random cropping/resizing. Meanwhile, the proposed method also exploits a symmetrized loss like DINO and MOCOv3 (pseudo code in Section G of Appendix), that’s **both the two image views are fed in teacher and student network**, thus there will be no problem of patch misalignment.
>
> [1] Dosovitskiy A, Beyer L, Kolesnikov A, et al. An image is worth 16x16 words: Transformers for image recognition at scale. arXiv preprint arXiv:2010.11929, 2020.

---

### Official Review · Reviewer_VUto · 2021-07-19

**Rating:** 6
**Confidence:** 4

**Summary:**

The authors propose a new framework for self-supervised learning from natural images. The proposed objective is a linear combination of two previously proposed objectives, namely student-teacher self-distillation (as in DINO), and masked-language modeling as is commonly used in NLP. The masking strategy is guided by the attention masks learned by the transformer backbone. The authors evaluate their model on standard image-SSL benchmarks: ImageNet classification and COCO detection and segmentation.

**Limitations And Societal Impact:**

Yes, the authors adequately addressed the limitations and potential negative societal impact of their work

**Main Review:**

The main weakness of the work is its novelty. The proposed objective simply adds a masked-language modeling loss to the objective in DINO. The question is whether the improvement upon DINO is worth the additional complexity, conceptual and otherwise. The improvement upon DINO is on the order of 1% for linear classification on ImageNet, detection on COCO, and segmentation on Cityscapes. This is fine in itself, the issue is that it is unclear whether there is much room to build on the work, as it is itself a simple combination of existing objective.

The author do highlight the importance of guiding the masking strategy with the learned attention masks however, showing it performs much better than random masking. This constitutes the main technical contribution of the paper, but wether this in itself justifies publication is unclear.

**Time Spent Reviewing:**

2

---

> ### Author Response · Authors · 2021-08-10
> **Response**
>
> Thank you for the positive comments and constructive feedback. Below are our responses to specific comments.
>
> **Question**: The main weakness of the work is its novelty.
>
> **Response**: In this paper, we point out two problems of current visual self-supervised learning:
>
> (a) the instance discrimination methods use the global features, but **lack of local information extraction**;
>
> (b) the masked language modeling (MLM) is prone to mask the tokens of crucial region for images but **lost spatial (global) information** (the MLM masks some tokens randomly and just predicts the original masked tokens).
>
> To overcome the above problems, in this paper, the instance discrimination method is imported from the theory of MLM. Our method can **capture the local relationships between patches while also understanding the global features**. The results of image-level task and dense-prediction tasks in Experiments 4 also show the effectiveness of our method. These results prove that our method is not a simply combination of the two existing methods.
>
> Specifically, our contributions mainly include:
>
> (a) We propose an attention-guided mask strategy to enhance the understanding of local context semantics in pre-training without damaging the crucial structure. As far as we know, this is the first work that deeply studies the MLM strategy in vision pre-training;
>
> (b) we propose to exploit a global decoder to further recover the spatial information, which greatly improves the versatility and scalability of the pre-training model.
>
> Here, we choose DINO as an example of the base model since it is a general and open source self-supervised framework. Moreover, the proposed method can be extended to the other frameworks.

---

> ### Author Response · Authors · 2021-08-31
> **MST is NOT a simple combination of two existing methods**
>
> Our further experiment shows that the simple combination, i.e., DINO + original MLM, only achieves 71.90% at 100 epochs. Please refer to the response to reviewer 2YYa. In fact, our MST contains two modules that the simple combination does not: 1) the attention-guided mask strategy, and 2) the global decoder. By introducing the attention-guided mask strategy, MST exploits a global decoder to restore the original image instead of the randomly masked tokens, extracting the crucial information of whole image. This crucial information is also used in the contrastive learning. The attention-guided mask strategy helps MST learn more representative and discriminative features through restoration and contrastive learning, achieving a considerable improvement in comparison to the simple combination. In experiments, our MST achieves accuracy 73.9%, higher 2% than that by using the simple combination.
>
> In summary, MST is NOT a simple combination of two existing methods. Rather,  it is a novel self-supervised learning method for vision tasks.

---

> > ### Comment · Reviewer_VUto · 2021-08-31
> > **Thank you for your response**
> >
> > After reading the authors' response to my and other reviewers' questions, I have upgraded my score from 5 to 6.

---

### Decision · Program_Chairs · 2021-09-27

**Decision:**

Accept (Poster)

**Comment:**


MST proposes to combine the task of mask language modeling with instance discrimination.

While adding non-negligible complexity to training, MST demonstrate gain of the order of 2% on various standard SSL tasks (ImageNet linear eval, MS-COCO, Cityscape). Additionally, reviewers all agreed that the ablation experiments performed during the rebuttal period clarified the importance of the contributions. Overall, it is unclear if the performance gain fully justifies the extra-complexity of the training pipeline.

Given the positive feedback from the reviewers and the good empirical results, I am in favor of acceptance